# Benchmarking Regridding Libraries Used in Earth System Modelling

**Sophie Valcke** *, **Andrea Piacentini and Gabriel Jonville**

Unité Mixte de Recherche 5318 «Climat Environnement Couplages et Incertitudes», Centre Européen de Recherche et Formation Avancée en Calcul Scientifique, Centre National de la Recherche Scientifique, CEDEX 1, 31057 Toulouse, France; piacentini@cerfacs.fr (A.P.); jonville@cerfacs.fr (G.J.)
* Correspondence: valcke@cerfacs.fr; Tel.: +33-(0)5-61-19-30-76

**Abstract:** Components of Earth system models (ESMs) usually use different numerical grids because of the different environments they represent. Therefore, a coupling field sent by a source model has to be regridded to be used by a target model. The regridding has to be accurate and, in some cases, conservative, in order to ensure the consistency of the coupled model. Here, we present work done to benchmark the quality of four regridding libraries currently used in ESMs, i.e., SCRIP, YAC, ESMF and XIOS. We evaluated five regridding algorithms with four different analytical functions for different combinations of six grids used in real ocean or atmosphere models. Four analytical functions were used to define the coupling fields to be regridded. This benchmark calculated some of the metrics proposed by the CANGA project, including the mean, maximum, RMS misfit, and global conservation. The results show that, besides a few very specific cases that present anomalous values, the regridding functionality in YAC, ESMF and XIOS can be considered of high quality and do not present the specific problems observed for the conservative SCRIP remapping. The evaluation of the computing performance of those libraries is not included in the current work but is planned to be performed in the coming months. This exercise shows that benchmarking can be a great opportunity to favour interactions between users and developers of regridding libraries.

**Keywords:** regridding; remapping; interpolation; Earth system modelling; code coupling; coupler; coupling library; coupled models; ocean-atmosphere general circulation models

## 1. Introduction

Component models assembled in Earth system models (ESMs) usually have different grids because of the different environments that they represent, e.g., in an ocean model, the latitude–longitude grid convergence singularity can be conveniently displaced over a continent. Therefore, the coupling fields sent by a source component model have to be transformed for use by a target component on its grid. The first step is to define the addresses and weights of the source grid points that will contribute to the calculation of the coupling field on the target grid. The second step is regridding, i.e., the multiplication of the source grid values by the regridding weights to express the coupling field on the target grid. This spatial transformation is called regridding, remapping, or interpolation.

Different libraries exist for regridding in ESMs, offering different algorithms. We briefly describe here the two-dimensional (2D) algorithms used. With a nearest neighbour algorithm, the values of the nearest neighbours on the source grid, possibly weighted by their distance to the target point, are associated to each target grid point. A first-order non-conservative approximation uses, for each target point, the values of the coupling field at the four enclosing source grid points, as in a bilinear algorithm. Different algorithms are implemented for higher-order (non-conservative) regridding: one widely used schema is the bicubic interpolation, which uses the values of the four enclosing source neighbours but also the values of the field gradients in each direction and the cross gradient in the

diagonal direction. In a first-order conservative remapping, the value for each target cell is computed as a weighted sum of the source cell values, with the contribution of a source cell being proportional to the fraction of the target cell intersected by the source cell. This method should be applied when it is important to conserve the area-integrated value of the coupling field, for example to conserve the energy associated with heat fluxes or water associated precipitation fields. The basis of a second-order conservative remapping is the same as the first-order conservative remapping but additional terms proportional to the gradients of the source field are applied.

The OASIS3-MCT (Ocean Atmosphere Sea Ice Soil 3—Model Coupling Toolkit) coupler [1] includes the SCRIP (Spherical Coordinate Remapping and Interpolation Package) library [2] for its regridding operations. A detailed analysis of the quality of the SCRIP library conservative remapping was realised in [3,4]. The impact of the different normalisation options and of a Lambert azimuthal projection above a certain latitude have been analysed for different types of grids. The general conclusion is that the SCRIP first-order conservative remapping may give satisfactory results for some types of grids for the different normalisation options; however, in some cases, only if the Lambert projection is activated and, in other cases, only if it is not. Furthermore, conservative regridding involving a Gaussian reduced grid always shows some problems, whether or not the Lambert projection is activated. This analysis motivated the exploration of other regridding libraries currently available for Earth system modelling, for a possible future interfacing in OASIS3-MCT. The regridding libraries analysed are the ones mostly used in Earth system modelling today, i.e., ATLAS, MOAB-Tempest Remap, YAC, ESMF and XIOS. The results of this exploration are presented in this paper and additional details can be found in [5]. Here we also show results for the SCRIP library, as a basis for comparison, but do not investigate specific problems when they arise, as the current objective is to evaluate alternative regridding libraries.

ATLAS [6] is an open-source library written in C++, currently being developed at the European Centre for Medium-Range Weather Forecast (ECMWF). It provides grids, mesh generation, and parallel data structures targeting numerical weather prediction or climate model developments. It is designed as an object-oriented modular library, with the capability to take advantage of the most recent computer architectures. It is meant to provide, among many other features, a set of parallel interpolation methods and is oriented toward the use of an internally consistent set of predefined grids and meshes. At the time of our evaluation, ATLAS provided nearest neighbour, linear, cubic and finite-element regridding methods but did not include any conservative remapping.

MOAB-Tempest Remap [7], which is also written in C++, is used in the energy exascale Earth system model (E3SM) [8], a state-of-the-art Earth system modelling project funded by the Department of Energy (DOE) in the United States. Through Fortran-compatible interfaces, it offers online conservative regridding based on a scalable advancing-front intersection algorithm, which allows to compute the supermesh defined by the intersection of the source and target grid cells. The supermesh is then used to assemble the higher-order, conservative, and monotonicity-preserving regridding weights.

YAC, Yet Another Coupler [9,10], is developed as a joint initiative between the German Climate Computing Center (DKRZ) and the Max Planck Institute for Meteorology (MPI-M). YAC is coded in C and a Fortran interface is also provided. Although targeting the German ICON (ICOsahedral Nonhydrostatic) model, the software provides multiple regridding methods, e.g., linear, nearest neighbour, first and second order conservative, and hybrid cubic Bernstein–Bézier patch [11] (see also Section 2.1.3) for the coupling of physical fields defined on regular and irregular grids on the sphere without a priori assumption about the particular grid structure or grid element types.

ESMF, the Earth System Modelling Framework [12,13], is an open-source software for coupling model components to form weather, climate, coastal, and other Earth science related applications. Today, ESMF is developed and governed by a set of partners in the USA that include the National Aeronautics and Space Administration (NASA), the National

Oceanic and Atmospheric Administration (NOAA), the U.S. Navy, the National Center for Atmospheric Research (NCAR) and the national Earth System Prediction Capability (ESPC). Using ESMF, the scientist only codes the scientific part of their model into modular components and adapts it to the standard calling interface and standard data structures of the framework. Different modules, coded by either the scientists themselves or by others, can then be assembled into large scientific applications. ESMF offers a full interface to Fortran 90 and partial interface to C/C++ and Python. The ESMF software provides the underlying layers necessary for an efficient parallel execution of the scientific applications on different computer architectures, allowing for the coupling of the module to other components. ESMF supports regridding on combinations of 2D or 3D, spherical or cartesian coordinates with different regridding methods: nearest neighbour, bilinear, higher order, based on patch algorithm (see Section 2.1.3), and first and second order conservative.

XIOS [14], standing for XML-IO-Server, is an open-source library written in C++ with a Fortran interface developed at the Institut Pierre-Simon Laplace (IPSL) and dedicated to the management of I/O in climate codes. XIOS offers an impressive ensemble of online operations on model data (file rebuilding, time series, seasonal means, regridding, vertical interpolation, compression, etc.) based on external XML metadata definition, in order to minimize the post-processing of the data. Its regridding utility offers first and second order conservative remapping (but no non-conservative algorithms) on any type of grids used in Earth system modelling. Recently, XIOS has also been used as a coupler, i.e., managing communication of data, not only between a component and a file, but also between two components.

In order to compare these libraries, several aspects have to be considered. In a preliminary analysis, we enquired about the available regridding methods and evaluated the general software development environment, e.g., the coding language, project history, development plans, provision of support to external projects, and committed manpower. This first analysis led us to conclude that ATLAS and MOAB-Tempest Remap are certainly appealing libraries with good long-term perspectives regarding their development and support. However, their usage for regridding in OASIS3-MCT cannot be recommended at this point, as some basic capabilities were still missing in the version evaluated (0.21), in particular the handling of missing/masked values for MOAB-Tempest Remap or conservative regridding for ATLAS [15].

Therefore, we pushed further the analysis for YAC, ESMF, and XIOS and decided to benchmark the quality of their regridding. We also analyzed SCRIP as a basis for comparison, using criteria proposed by Coupling Approaches for Next-Generation Architectures (CANGA) project [16]. CANGA is a joint effort funded by the United States Department of Energy's Office of Science under the Scientific Discovery Through Advanced Computing (SciDAC) program that targets new high-performance coupling approaches for Earth system models on next-generation computers. Following CANGA, aspects to consider when evaluating a regridding library are the sensitivity (i.e., the algorithmic invariance of the scheme to the underlying mesh topology), the global conservation of integral quantities, the consistency (i.e., the preservation of discretization order and accuracy), the monotonicity (i.e., the preservation of global solution bounds), the dissipation (i.e., the smoothing of local solution maxima and minima that has to be minimal), the scalability, and the performance of the library. CANGA proposes metrics to quantify these aspects and we implemented the calculation of some of these metrics in our benchmark. The benchmark characteristics are detailed in Section 2.1, while its specific use for evaluating SCRIP, YAC, ESMF and XIOS is described in Section 2.2. In Section 3, we detail the benchmark results obtained for the four libraries. Finally, conclusions and perspectives of this work are presented in Section 4.

## 2. The Regridding Benchmark

Here, in Section 2.1, we describe the characteristics of the benchmark used to evaluate the regridding libraries that includes five algorithms, four different functions, and different combinations of six grids used in real ocean or atmosphere models. In Section 2.2, we provide some details on its application for the four libraries SCRIP, YAC, ESMF and XIOS.

### 2.1. The Benchmark Characteristics

#### 2.1.1. Grids

The six grids considered in the benchmark are the following, given with their acronym used in the rest of the document and number of grid points:

- *torc*: the ocean model NEMO (Nucleus for European Modelling of the Ocean) [17], rotated-stretched logically-rectangular grid with $182 \times 149$ points horizontally;
- *nogt*: the ocean model NEMO, rotated-stretched logically-rectangular grid with $362 \times 294$ points horizontally;
- *bggd*: the atmosphere model LMDz (Laboratoire de Météorologie Dynamique zoom), [18] regular latitude–longitude grid with $144 \times 143$ points horizontally;
- *sse7*: the atmosphere model ARPEGE (Action de Recherche Petite Echelle Grande Echelle) [19], Gaussian reduced T127 with 24,572 points horizontally (unstructured, described with up to 7 vertices per cell);
- *icos*: the atmosphere model Dynamico [20], low-resolution unstructured icosahedral grid with 15,222 points horizontally;
- *icoh*: the atmosphere model Dynamico, high-resolution unstructured icosahedral grid with 2,016,012 points horizontally.

These grids are illustrated on Figure 1.

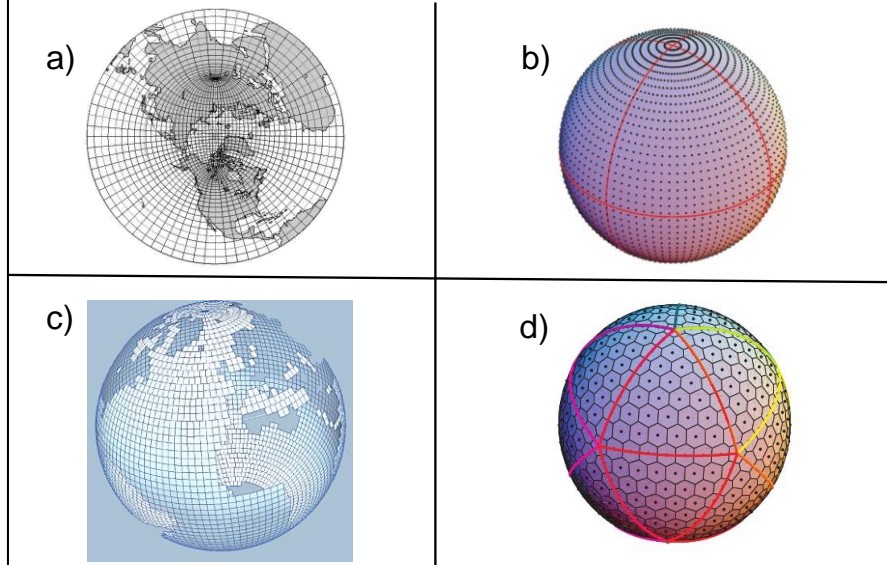

**Figure 1.** Illustration of the types of grids included in the benchmark: (**a**) rotated-stretched logically-rectangular (*torc, nogt*), (**b**) regular latitude–longitude (*bggd*), (**c**) Gaussian-reduced (*sse7*), and (**d**) icosahedral (*icos, icoh*).

The first five grids are relatively low-resolution grids. We decided to run the benchmark for the six pairs of these grids matching an ocean and an atmospheric grid and introduced the higher-resolution *icoh* grid only to test the impact of large resolution differences on the conservative regridding.

We note here that all grids used in this benchmark define a sea-land mask, with valid (non-masked) points over the ocean and not valid (masked) point over the land. In order to avoid non-matching sea-land masks between the ocean and the atmosphere grids, we adopted the following best practice that sets up a consistent atmosphere-ocean system and defines a well-posed coupled problem: The original sea-land mask of the ocean model is taken as is. For the atmosphere model, the fraction of water in each cell is defined by the conservative remapping of the ocean mask on the atmospheric grid. Then, the atmospheric coupling mask is adapted by associating a valid/active index to cells containing at least a surface fraction 1/1000 of water. Under 1/1000 of water, the atmospheric cell is considered

to be completely masked. This method ensures that the total sea and land surfaces are the same in the ocean and atmosphere models, allowing global conservation of sea or land integrated quantities. It also minimizes the number of target grid points that does not receive a value with each specific regridding algorithm.

### 2.1.2. Analytical Functions

The four analytical functions used to define the coupling fields to be regridded, illustrated on Figure 2, are (see also Appendix A for their exact definition expressed in Fortran 90):

(a)  *sinusoid*: a slowly varying standard sinusoid over the globe;
(b)  *harmonic*: a more rapidly varying function with 16 maximums and 16 minimums in northern and southern bands;
(c)  *vortex*: a slowly varying function with two added vortices, one in the Atlantic and one over Indonesia;
(d)  *gulfstream*: the slowly varying standard sinusoid with a mimicked Gulf Stream.

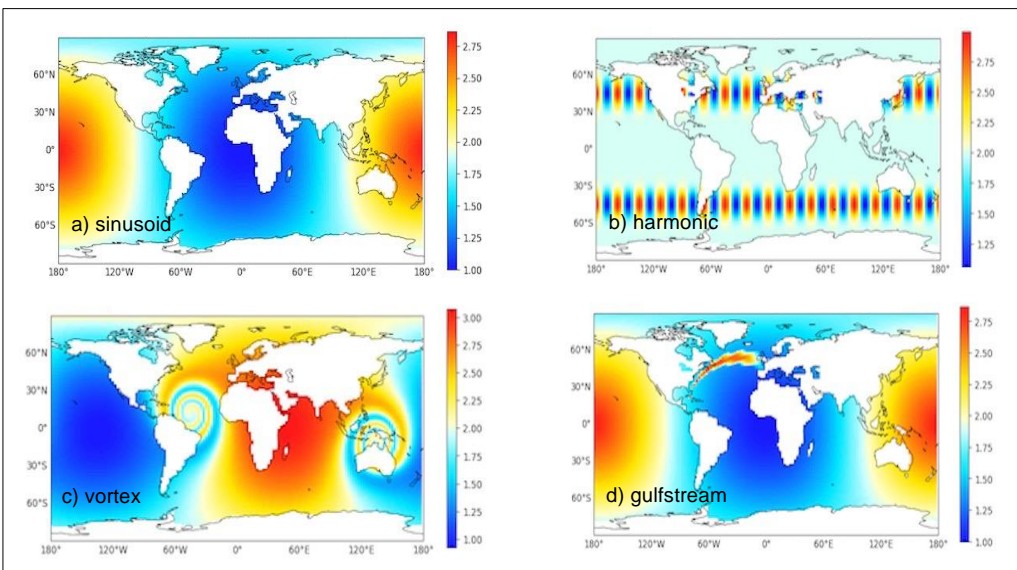

**Figure 2.** The four analytical functions defining the coupling field: (**a**) *sinusoid*, (**b**) *harmonic*, (**c**) *vortex*, (**d**) *gulfstream*.

### 2.1.3. Regridding Algorithms

The following algorithms were evaluated for the different regridding libraries, when available. The particularities of the algorithm for each library are described. We also specifically mention the option activated in the different regridding libraries to ensure that all valid target grid points receive a regridded value, even near the coasts.

1.  Nearest neighbour

For all libraries, except for XIOS, which does not implement this algorithm, the value of the non-masked nearest neighbour on the source grid was assigned to each target grid point, i.e., only one neighbour was used.

For ESMF, the options allowing regridding on the cell centre locations of an unstructured grid (i.e., *–src_loc center –dst_loc center*) and the option ignoring degenerate cells in either the source or the destination grid (*–ignore_degenerate*) were activated. This option can be useful for the NEMO grids *torc* and *nogt*, which may have masked cells (i.e., not used in the regridding) collapsing into a point or line.

2. First order non-conservative

SCRIP uses a general scheme based on a local bilinear approximation. For non-masked target points that do not receive a value with the original bilinear algorithm, as can happen near the coast, the nearest non-masked source neighbour value was used, by default.

ESMF uses a standard bilinear algorithm. The same options as for the nearest neighbour regridding were activated (i.e., *–src_loc center –dst_loc center, –ignore_degenerate*). In addition, the option *–extrap_method neareststod* is turned on. Each target point that did not receive a value with the original algorithm used the closest unmasked source point to define its value (in order to reproduce the default behaviour of the SCRIP library).

We also note that with ESMF, grids can be described with the so-called *SCRIP* format or with an *unstructured* format. The *SCRIP* format (not to be confused with the SCRIP library itself) describes the grid with the latitude and the longitude of the centre and corners of each cell. The *unstructured* format describes the grid as an ensemble of elements and provides the element connectivity associating for each element a certain number of nodes in the list of nodes for which the latitude and longitude are provided.

For YAC, we activated an inverse-distance weighting of the vertex values of the source polygon enclosing the target point, and an average of the two nearest neighbours for target points falling outside any source polygon, so to ensure that all non-masked target grid points receive a regridded value.

XIOS does not implement any first-order non conservative regridding.

3. Second order non-conservative

For SCRIP, the bicubic regridding follows the general local bilinear remapping using the values of each vertex of the enclosing source cell and the values of the gradients in each local direction and in the cross direction. Again, the nearest non-masked source neighbour value is used for non-masked target points that do not receive any value with the original bicubic algorithm.

For YAC, the recently introduced hybrid cubic spherical Bernstein–Bézier (HCSBB) method [11] was used [10]. Compared to the patch algorithm used in ESMF (see below), the HCSBB method always results in an interpolated field that has a continuous first derivative. The source grid was first triangulated and the derivatives of the source field across the edges of the triangles were computed. Triangular patches were constructed from a blend of spherical Bernstein–Bézier polynomials using these derivatives, and then used to regrid each target point. Compared to the patch algorithm, this method uses a bigger stencil to compute each target point. The completion with 4-nearest non-masked neighbours is also activated for non-masked target points that do not receive any value with the original HCSBB algorithm.

For ESMF, the patch algorithm that is used is a technique commonly used in finite element modelling. Patch interpolation works by constructing multiple polynomial patches for the cells around the vertices of a source cell (e.g., for a square source cell four patches would be computed). For 2D grids, these polynomials are currently second degree 2D polynomials. The interpolated value at the destination point is the weighted average of all the patches for the source cell (e.g., the four patches for a square cell). This patch averaging prevents too strong overshoots and undershoots. The same options as for the first order non-conservative regridding (i.e., *–src_loc center –dst_loc center –ignore_degenerate –extrap_method neareststod)* were activated.

XIOS does not implement any second order non-conservative regridding.

4. First order conservative with FRACAREA and DESTAREA normalisations

In a first-order conservative remapping, the value for each target cell is computed as a weighted sum of source cell values, with the contribution of a source cell being proportional to the fraction of the target cell intersected by the source cell. In case of non-matching sea-land masks between the atmosphere and the ocean grids, different normalisation options exist. DESTAREA (DESTination AREA) uses the whole target cell area for the normalisation, whereas FRACAREA (FRACtional AREA) uses the intersected area of

the target cell. DESTAREA ensures local conservation but may produce non-physical values while FRACAREA does not ensure local conservation but produces values that are physically consistent. We note also that the FRACAREA normalisation may give some good results for the wrong reasons, in the sense that the normalisation operation involving the intersected target cell area, as calculated by the library itself, may lead to the cancellation of error present in the weights before the normalisation. DESTAREA does not involve this error cancellation and therefore often reveals specific algorithmic problems. All libraries implement both normalisation options.

For conservative remappings, the SCRIP library assumes by default that the edges of the meshes follow a straight path in the longitude–latitude space. It is however possible, for the edge intersection calculation, to switch to a Lambert equivalent azimuthal projection above a certain latitude threshold if specified. We performed the benchmark tests either without any projection, or with a projection above 1.45 radians in latitude north. In the latter case, the results are denoted as *SCRIP-L* and in the former case, they are denoted as *SCRIP*. We mention here that, by default, target cells that do not intersect any non-masked source cells do not receive any value, even if this never happens in our tests thanks to the approach use to define the sea-land masks (see Section 2.1.1).

For conservative remapping, ESMF assumes by default that grid cells edges follow great circle paths along the sphere surface. The default normalisation is DESTAREA. To activate the FRACAREA normalisation, the option *–norm_type fracarea* was activated. The option *–ignore_degenerate* (see above) was also activated. In addition, the option *–ignore_unmapped*, i.e., do not do anything special for target point that does not receive a value with the original algorithm, was activated in order to reproduce the default behaviour of SCRIP.

With XIOS, the mesh edges can be described with great circle or latitude circles, and is automatically defined by the grid type. For unstructured and curvilinear grids (i.e., *torc*, *nogt, icos,* and *icoh* in our case), great circles are used. For longitude–latitude (i.e., *bggd* in our case), and Gaussian-reduced (i.e., *sse7*), latitude circles are used for the edges located on a latitude circle and great circles are used otherwise.

With YAC, the edges of the grid cells can be either defined with longitude and latitude circles or with great circles depending on the interface used. We used the interface defining the edges of the grid cells with great circles. We have to note here that this is not totally appropriate for the cell edges following a latitude circle as in the regular latitude-longitude grid *bggd* and in the Gaussian-reduced grid *sse7*.

5. Second order conservative with FRACAREA normalisation

As stated above, the basis of a second-order conservative remapping is the same as for the first-order conservative remapping but additional terms proportional to the gradients of the source field are applied. While remaining conservative, this remapping ensures that field details are reconstructed and that different target cells entirely located under the same source cell receive different values. This difference between the first-order and second-order methods is particularly apparent when going from a coarse source grid to a finer destination grid (see Section 3.6). Another difference is that the second-order method does not guarantee that after regridding the range of values in the destination field is within the range of values in the source field. For example, if the minimum value in the source field is 0.0, it is possible that after regridding the destination field contains negative values.

SCRIP applies gradients calculated in the longitudinal and latitudinal directions.

YAC, ESMF, and XIOS implement the second-order conservative algorithm based on [21]. For all four libraries, in cases where the gradient computation fails (for example due to a lack of neighbours, which can occur at land-sea mask borders), the algorithm automatically assumes a zero gradient, which is essentially a fall back to a first-order conservative remapping.

For ESMF, the same options used for the first-order conservative remapping (i.e., *–ignore_unmapped –ignore_degenerate*, and *–norm_type fracarea*) were activated.

2.1.4. Benchmark Metrics

The benchmark implements the calculation of regridding metrics proposed by the CANGA project. With the following definitions:

- $\Psi^s$: the analytical function on the source grid;
- $\Psi^t$: the analytical function on the target grid;
- $R\Psi^s$: the source analytical function regridded on the target grid;
- $I_s$: the integral on the source grid;
- $I_t$: the integral on the target grid;

The CANGA metrics are defined as:

- mean misfit: mean $(|R\Psi^s - \Psi^t|/|\Psi^t|)$;
- maximum misfit: max $(|R\Psi^s - \Psi^t|/|\Psi^t|)$;
- RMS (root mean square) misfit: RMS $(|R\Psi^s - \Psi^t|/|\Psi^t|)$;
- $L_{min}$: $(\min \Psi^t - \min R\Psi^s)/\max (|\Psi^t|)$ (A positive $L_{min}$ detects an overestimate of the function minimum (i.e., it reinforces the minimum) while a negative $L_{min}$ detects some smoothing of the function minimum);
- $L_{max}$: $(\max R\Psi^s - \max \Psi^t)/\max (|\Psi^t|)$ (A positive $L_{max}$ detects an overestimate of the function maximum (i.e., it reinforces the maximum) while a negative $L_{max}$ detects some smoothing of the function maximum);
- Source global conservation: $|I_t (R\Psi^s) - I_s (\Psi^s)|/I_s (\Psi^s)$;
- Target global conservation: $|I_t (R\Psi^s) - I_t (\Psi^t)|/I_t (\Psi^t)$.

We calculated these metrics for all libraries for all pairs of grids for the 4 functions for all algorithms except when the library did not support the algorithm.

### 2.2. Implementation of the Regridding Benchmark for SCRIP, YAC, ESMF and XIOS

The steps to realize in order to calculate the benchmark metrics for each regridding library is, of course to download the library sources, compile them, and develop a scripting environment to generate regridding weights activating the different regridding algorithms for the different pairs of grids. We went through these steps for YAC, ESMF and XIOS. For completeness, we also describe the environment used to generate the weights with the SCRIP library, as the benchmark metrics were also calculated for the SCRIP for comparison. These calculations were realized by different developers on different platforms, using the intel 18.0.1.163 compiler and associated intel mpi 2018.1.163. The current benchmark results, evaluating the quality of the regriddings, are not sensible to the platform used, while a benchmark evaluating the numerical performance of the libraries would be.

- SCRIP

The OASIS3-MCT, and therefore SCRIP, sources used for the regridding benchmark correspond to the trunk of the OASIS3-MCT git developer repository dated 05/05/2021. The environment used to calculate regridding weights with the SCRIP library in OASIS3-MCT is available on Zenodo (see the Data Availability section below). The benchmark tests were run on LENOVO cluster nemo at CERFACS (288 bi-socket nodes with 12 Intel cores E5-2680-v3 2.5 Ghz with 64 GB of memory).

- ESMF

The sources used for the results presented in Section 3 correspond to the branch ESMF_8_2_0_beta_snapshot_08. An environment developed to generate regridding weights with ESMF is available on Zenodo. As for SCRIP, the benchmark tests were run on LENOVO cluster nemo at CERFACS.

- YAC

YAC sources used for the regridding benchmark corresponds to a pre-release state of YAC v2.0.0 that was provided by the developers. All developments used in this version are now included in the official release YAC v2.3.0. The environment to calculate regridding

weights with YAC is available on Zenodo. All regridding weight calculations were done on a PC Dell Precision M7720 with 6 cores Intel Xeon E-2186M, 64 Gb RAM.

- XIOS

The sources used for the results presented in Section 3 correspond to SVN revision 2134 dated 2021-04-29. The environment developed to generate regridding weights with XIOS is available on Zenodo. As for YAC, all regridding weight calculations were done on a PC Dell Precision M7720 with 6 cores Intel Xeon E-2186M, 64 Gb RAM.

Once the regridding weights had been generated, the benchmark metrics were calculated for the four libraries using different analytical functions using a specific scripting environment based on Python 3.7.7 available on Zenodo.

## 3. Benchmark Results

All benchmark metrics were calculated for:

- the four analytical functions: *sinusoid, harmonic, vortex, gulfstream* (see Section 2.1.2);
- the six pairs of relatively low-resolution grids matching an ocean grid with an atmospheric grid: *torc-bggd, torc-icos, torc-sse7, nogt-bggd, nogt-icos, nogt-sse7* in both directions (see Section 2.1.1); for the conservative remapping, we also analyse the regridding of the *vortex* function for *icos-icoh* and *nogt-icoh* in order to test the impact of that regridding on cases with large resolution difference (see Section 3.6);
- for the four regridding libraries: SCRIP (+SCRIP-L, i.e., with Lambert projection for conservative regridding), YAC, ESMF and XIOS;
- for all algorithms: nearest neighbour, 1st and 2nd order non-conservative, 1st and 2nd order conservative, except when the regridding library does not support the algorithm, such as, e.g., nearest neighbour for XIOS (see Section 2.1.3).

Results of all metric values and plots are available on Zenodo. The lists of the individual files containing metric values and plots are detailed in Appendices B and C, respectively.

We analysed all metrics obtained but we cannot of course discuss them all here. In the next paragraphs, we present specific cases, either to illustrate the main conclusions of our analysis or to highlight the specific problems observed. We note here that we show metric results for the SCRIP library, as a basis for comparison. However, if specific problems are revealed by the benchmark for the SCRIP, we do not further investigate them as the current objective is to evaluate other regridding libraries.

### 3.1. Nearest Neighbour Regridding

Figure 3 shows the mean, rms and maximum misfit for the different pairs of grids for the *harmonic* function for the nearest neighbour regridding. The three regridding libraries produce almost exactly the same, and very reasonable, results: the curves are superimposed and not distinguishable. This is also true for the other analytical functions (not shown).

We observed that the function used to define the coupling field has a strong impact on the maximum misfit, as illustrated on Figure 4, which shows the maximum misfit for the different pairs of grids for the four functions. The maximum misfit is directly linked to the gradient of the function, being much higher for example for the *gulfstream* function than for the slowly varying *sinusoid* function, as is expected for a nearest neighbour algorithm.

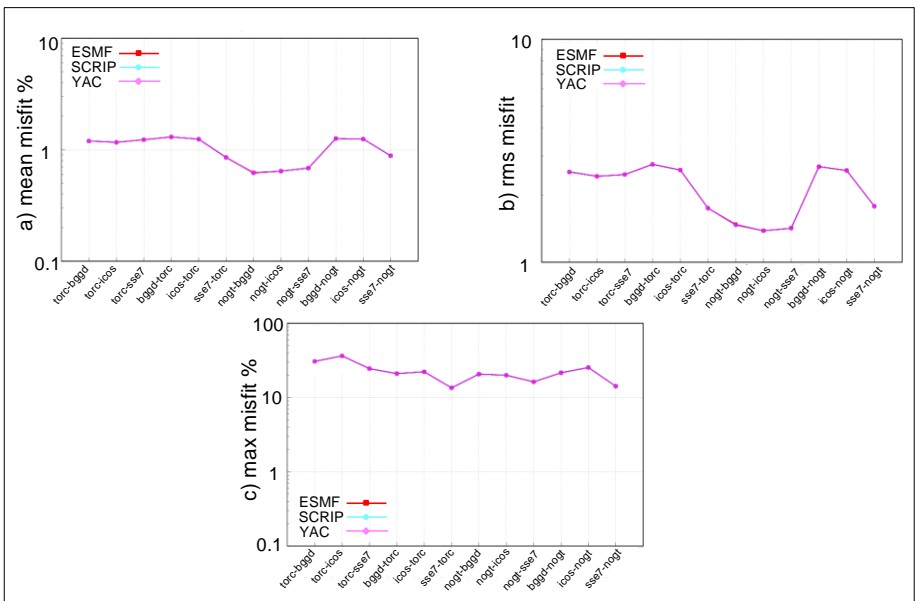

**Figure 3.** (**a**) mean, (**b**) rms and (**c**) maximum misfit for the different pairs of grids for the *harmonic* function for nearest neighbour algorithm for ESMF, SCRIP and YAC.

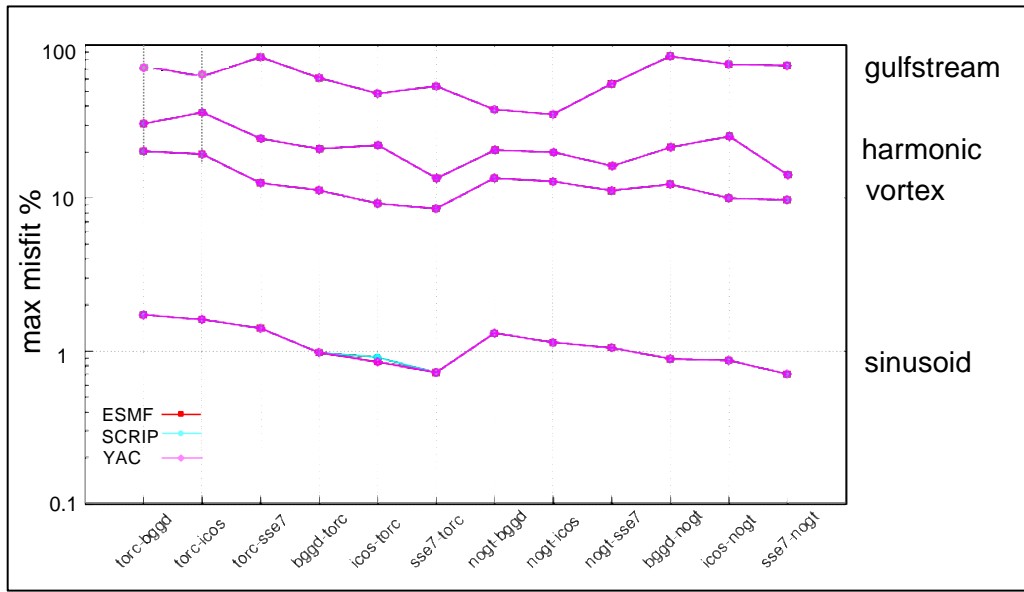

**Figure 4.** Maximum misfit for the different pairs of grids for the different functions *sinusoid, vortex, harmonic, gulfstream* for the nearest neighbour algorithm for ESMF, SCRIP and YAC.

### 3.2. 1st Order Non-Conservative Regridding

Figure 5 shows the mean, rms, and maximum misfit for the different pairs of grids for SCRIP, ESMF and YAC, for the *vortex* function for the first-order non-conservative regriddings described in Section 2.1.3. The algorithm in YAC is less accurate on average, i.e., the mean misfit is higher on average. This was also observed for the other analytical functions (not shown).

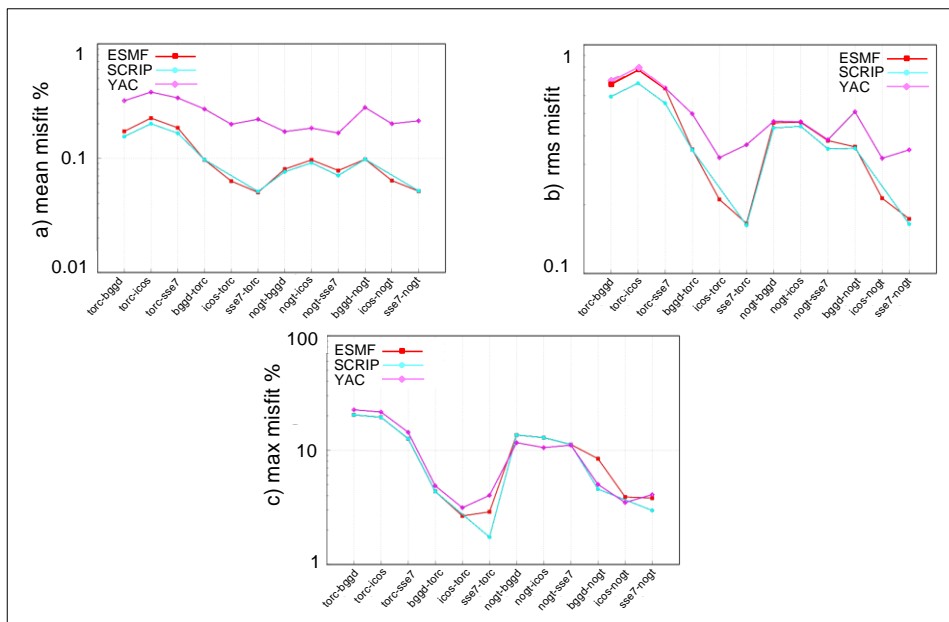

**Figure 5.** (**a**) mean, (**b**) rms, and (**c**) maximum misfit for the different pairs of grids for the *vortex* function for first-order non-conservative regridding for ESMF, SCRIP and YAC.

### 3.3. Second-Order Non-Conservative Regridding

Second-order non-conservative algorithms are available in SCRIP, ESMF and YAC (see details in Section 2.1.3). Figure 6 shows the mean misfit, rms misfit, maximum misfit, and $L_{max}$ for the different pairs of grids for these three regridding libraries for the *gulfstream* function.

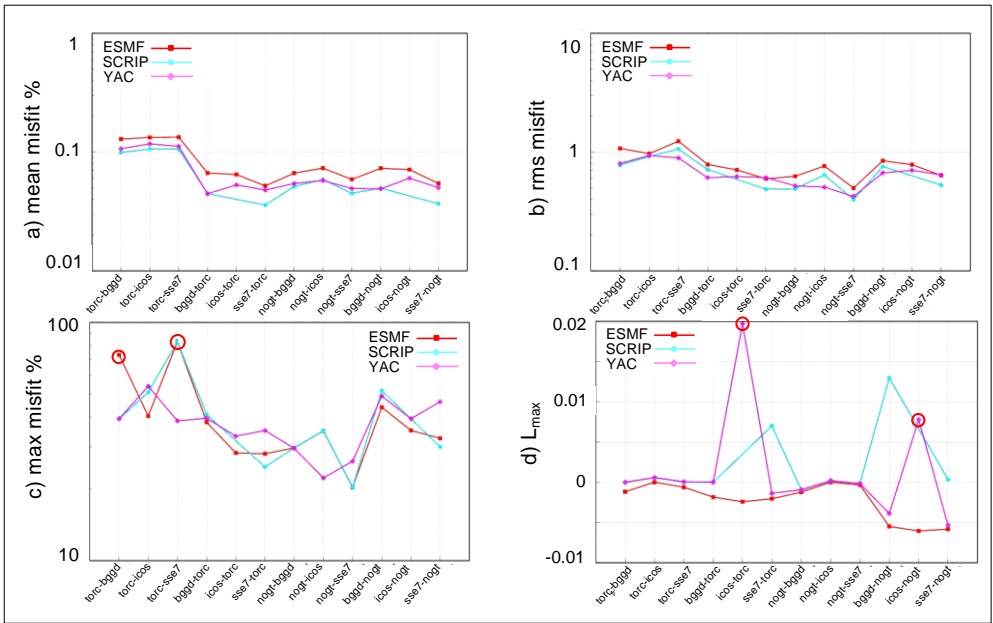

**Figure 6.** (**a**) mean, (**b**) rms, and (**c**) maximum misfit and (**d**) $L_{max}$ for the different pairs of grids for the *gulfstream* function for second-order non-conservative algorithms for ESMF, SCRIP and YAC. The red circles identify anomalous regriddings detailed in the text.

On average, the SCRIP bicubic algorithm gives slightly better results for certain pairs of grids and the ESMF patch algorithm gives slightly less accurate results (Figure 6a). The averaging present in the ESMF patch algorithm smooths the regridded field and prevents overshoots or undershoots, as can be seen by the more negative values for $L_{max}$.

In Figure 6c, we note some high maximum misfit for ESMF for *torc-bggd* and *torc-sse7*, not present for the other functions (not shown). These anomalous points also appear for the bilinear regridding for the *gulfstream* function only (not shown). This led us to look for anomalous regridded values in the gulf stream region. The 2D plots of the misfit in that region for the *gulfstream* function for the *torc-bggd* regridding are shown at Figure 7. One anomalous value near the coast (in yellow) is indeed easy to identify for ESMF patch algorithm. The same anomalous point appears for the *torc-sse7* regridding (not shown). At the time of writing this paper, this particular case was under investigation with ESMF developers.

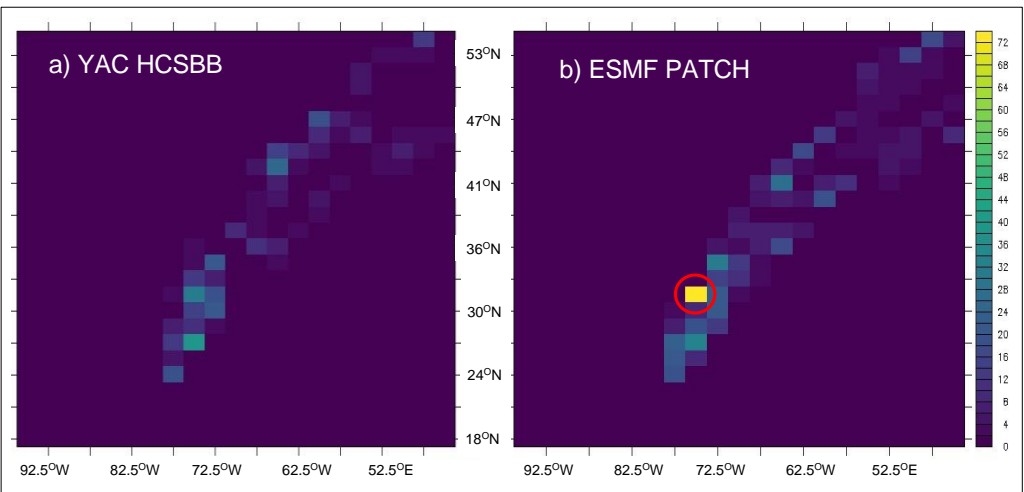

**Figure 7.** Misfit (%) for the *gulfstream* function in the gulf stream region for *torc-bggd* regridded with the second-order non-conservative algorithm for (**a**) YAC HCSBB and (**b**) ESMF PATCH. The red circle identifies the anomalous value near the coast for the ESMF patch algorithm discussed in the text.

Figure 6d also shows high values of $L_{max}$ for *icos-torc* and *icos-nogt* for the *gulfstream* function that do not appear for the other functions (not shown). Figure 8 shows 2D plots of the regridded field in the gulf stream region for ESMF and YAC. Indeed, it confirms that, compared to ESMF, which tends to smooth the local maximum with its patch averaging algorithm, YAC gives higher, but a priori non-anomalous, values in the centre of the gulf stream.

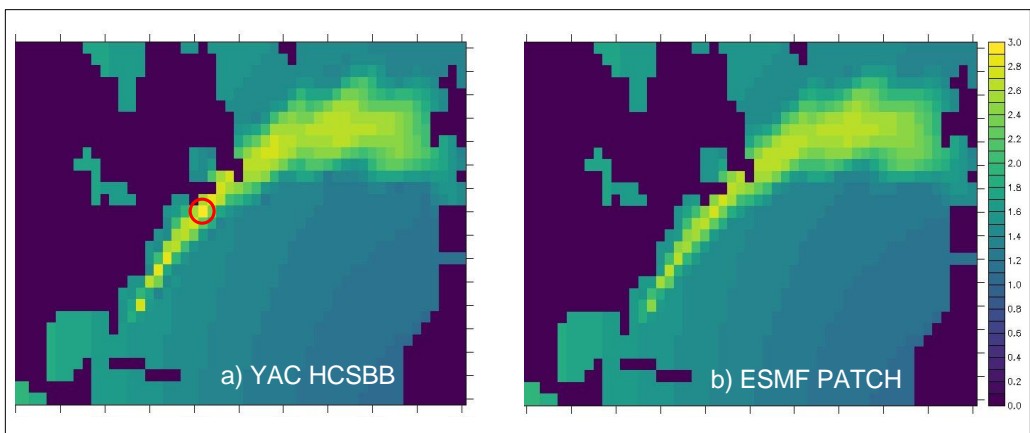

**Figure 8.** Misfit (%) for the *gulfstream* function in the gulf stream region for *icos-torc* regridded with the second-order non-conservative algorithm for (**a**) YAC HCSBB and (**b**) ESMF PATCH. The red circle identifies the highest, but a priori non-anomalous, value in the centre of the gulf stream for YAC discussed in the text.

### 3.4. First-Order Conservative Remapping with DESTAREA Normalisation

To evaluate the quality of the first-order conservative regridding, we started by looking at the results obtained with the DESTAREA normalisation option, which usually reveals

problems that the FRACAREA option would hide, sometimes involving a cancellation of errors. Figure 9 shows the mean and the maximum misfits for the *harmonic* function for the four libraries. Here, for ESMF, *nogt* and *torc* are described with the *unstructured* grid format (see Section 2.1.3). It confirms the extremely wrong values obtained using the SCRIP library either activating (SCRIP-L) or not activating (SCRIP) the Lambert azimuthal projection, as mentioned in the introduction (see also [4,5]). The other libraries ESMF, YAC and XIOS produced practically the same and satisfactory results, with a mean misfit between 0.1% and 1% and a maximum misfit between 1% and 10% for all pairs of grids.

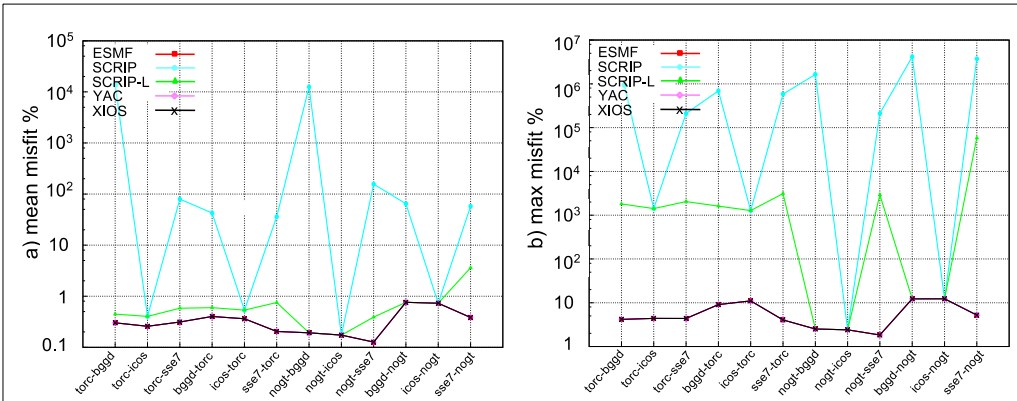

**Figure 9.** (**a**) mean and (**b**) maximum misfit for ESMF, SCRIP, SCRIP-L, YAC, and XIOS for the first-order conservative remapping with DESTAREA normalisation for the different pairs of grids for the *harmonic* function. For ESMF, *nogt* and *torc* are described with the *unstructured* format.

Figure 10 shows the source global conservation metric for the 4 functions for all regridding libraries for the different pairs of grids. Again, it is very clear that the SCRIP/SCRIP-L library presents some important problems with the first-order conservative remapping. On the contrary, ESMF, YAC, and XIOS show similar and very reasonable results, this metric being at maximum of the order of 1%.

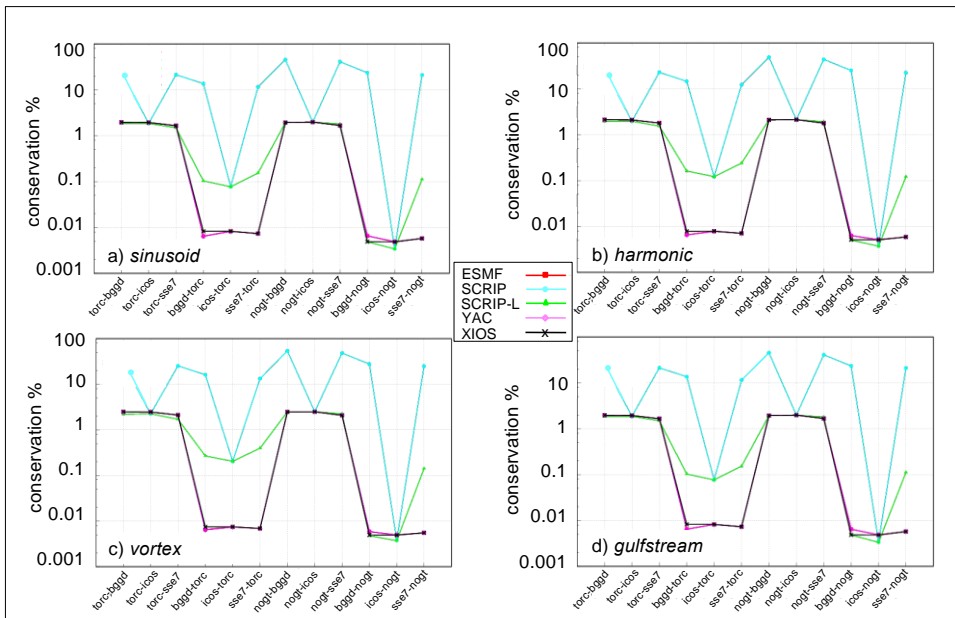

**Figure 10.** Source global conservation (%) for ESMF, SCRIP, SCRIP-L, YAC and XIOS for the 1st order conservative remapping with DESTAREA normalisation for the different pairs of grids for the 4 functions: (**a**) *sinusoid*, (**b**) *harmonic*, (**c**) *vortex*, and (**d**) *gulfstream*.

We then analysed the impact of the grid description format in ESMF. As explained in Section 2.1.3 two formats are supported to describe the grids with ESMF, either the so-called *SCRIP* format or the *unstructured* format. The results above were produced describing the ocean NEMO grids *nogt* and *torc* with the *unstructured* format. However, the *nogt* and *torc* grids are structured, and it is possible to describe them using the *SCRIP* format. As such, we repeated the first-order conservative regriddings for ESMF using the *SCRIP* format to describe the *nogt* and *torc* grids. Figure 11 shows the mean and the maximum misfit for the *harmonic* function in that case. The results are the same as on Figure 9, except that ESMF now presents anomalous mean and maximum misfits when *nogt* is the source grid. The same anomalies are observed for the other functions (not shown).

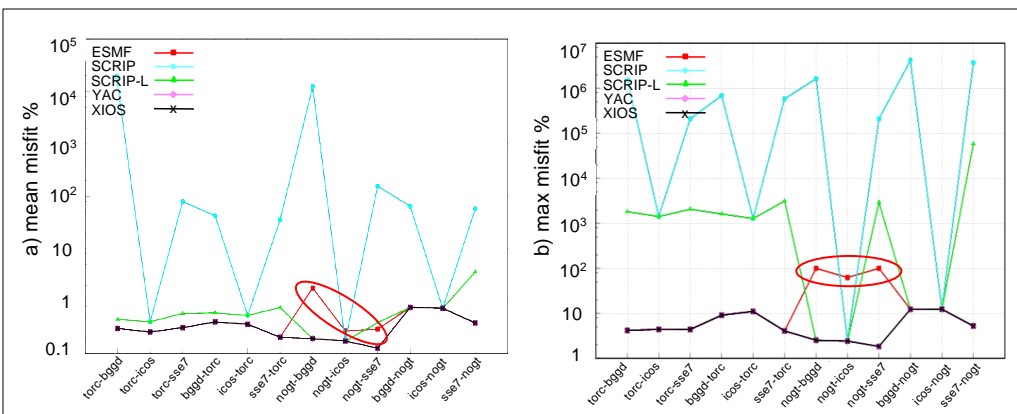

**Figure 11.** (**a**) Mean and (**b**) maximum misfit for ESMF, SCRIP, SCRIP-L, YAC, and XIOS for the first-order conservative remapping with DESTAREA normalisation for the different pairs of grids for the *harmonic* function. For ESMF, *nogt* and *torc* are described with the *SCRIP* structured format. The red oval shapes identify ESMF regriddings showing anomalous mean and maximum misfits when *nogt* is the source grid. These regriddings are discussed in the text.

Figure 12 shows the 2D plot of the misfit of the *harmonic* regridded function for *nogt-bggd* with *nogt* described (a) with the *SCRIP* format and (b) with the *unstructured* format. The problem, clearly linked to the north fold of the NEMO, disappears when *nogt* is described with the *unstructured* format.

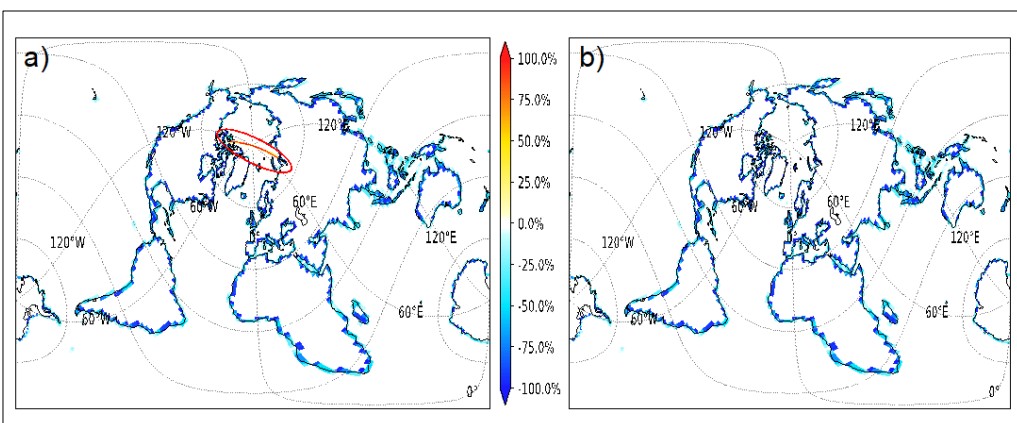

**Figure 12.** Two-dimensional plot of the misfit of the *harmonic* function regridded with ESMF first-order conservative remapping for *nogt-bggd* with *nogt* described (**a**) with the *SCRIP* format and (**b**) with the *unstructured* format. The red oval shape identifies in (**a**) the grid points linked to the north fold of the NEMO grid showing anomalous misfit when the *nogt* grid is described in the SCRIP format.

It is interesting to note that the regridding does not show any specific problem when *torc* is the source grid. This is certainly linked to the type of grid in the north fold. For *torc*, the north fold is such that in the (i,j) space the third-to-last row folds on the last row and the penultimate row folds on to itself. For *nogt*, the penultimate row folds on the last row. As for the anomaly identified for the patch regridding for the *gulfstream* function (see Section 3.3), this problem is, at the time of writing, under investigation with ESMF developers.

### 3.5. 1st Order Conservative Remapping with FRACAREA Normalisation

Figure 13 shows the maximum misfit for the first-order conservative regridding with FRACAREA normalisation for the four functions for all pairs of grids. For ESMF, the *nogt* grid is described with the *unstructured* format to avoid specific problems linked to the north fold (see Section 3.4). All regridding libraries have the same maximum misfit, except SCRIP and SCRIP-L, which we will not further discuss here. As expected, the maximum misfit is higher for the functions with sharper gradients. For example, the maximum misfit is higher for the *harmonic* function than for the *sinusoid* function for all pairs of grids. For the *gulfstream* function (Figure 13d), the maximum misfit for *torc-sse7* is particularly high. As this is the case for all regridding libraries and not for the other functions, this is probably linked to the sharp gradients of the *gulfstream* function.

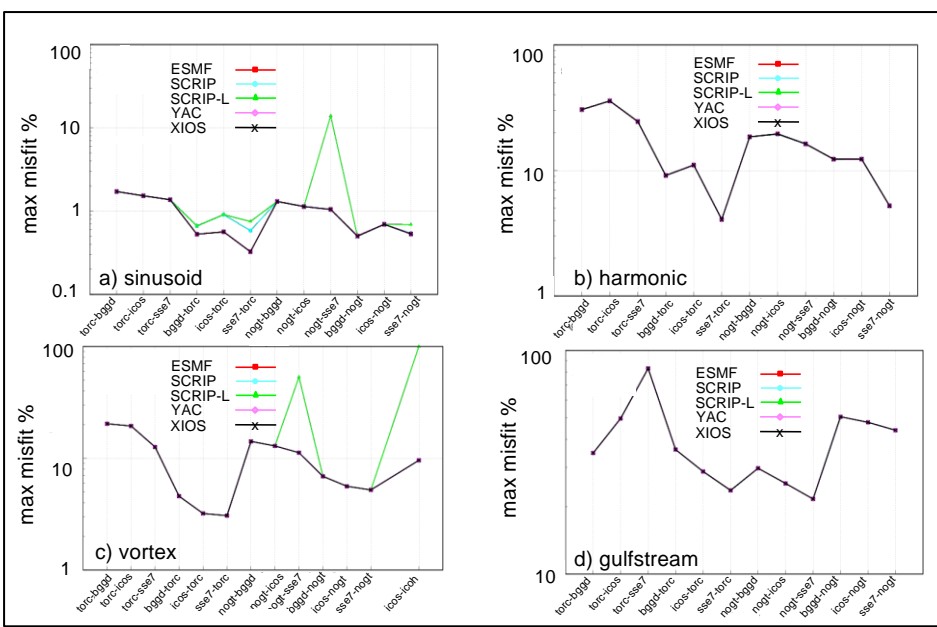

**Figure 13.** Maximum misfit for ESMF, SCRIP, SCRIP-L, YAC, and XIOS for the first-order conservative remapping with FRACAREA normalisation for the different pairs of grids for the 4 functions: (**a**) *sinusoid*, (**b**) *harmonic*, (**c**) *vortex*, (**d**) *gulfstream*.

For the source global conservation metric (not shown), ESMF, YAC, and XIOS show similar and very good results, this metric being less than 0.01% in all cases. The source global conservation for the *icos-icoh* pair of grids for the *vortex* function, also calculated for that regridding, is remarkably small, being of the order of $10^{-9}$.

### 3.6. Second-Order Conservative Remapping with FRACAREA Normalisation

Figure 14 shows the mean, maximum, rms misfits, and the source global conservation for the second-order conservative remapping with the FRACAREA normalisation for the different pairs of grids for all regridding libraries for the *harmonic* function. Besides SCRIP and SCRIP-L, which we will not further analyse here, we see that all regridding libraires show more or less the same behaviour with good global conservation. This is not surprising, as they all implement the same algorithm based on [21].

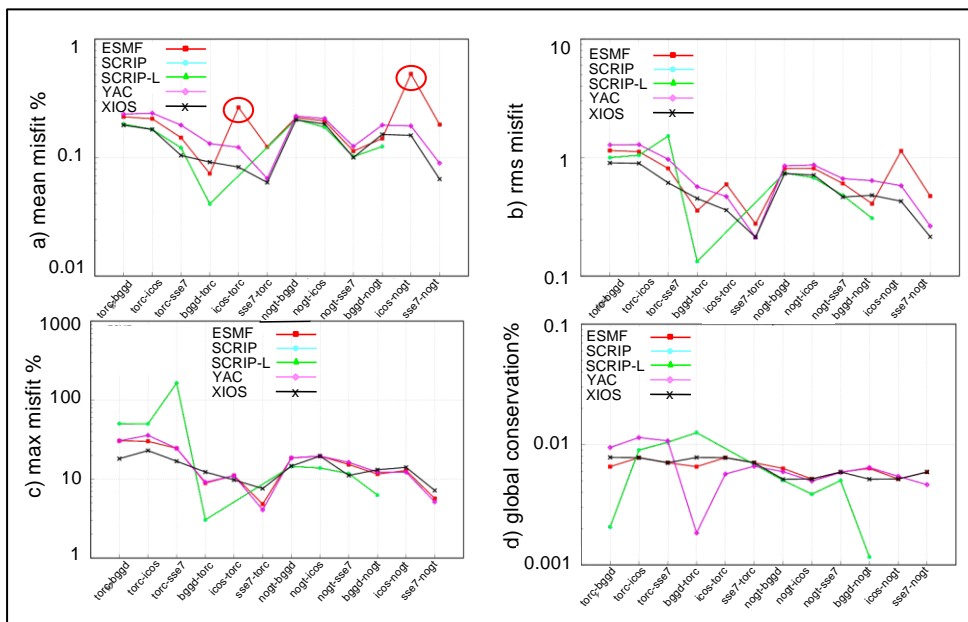

**Figure 14.** (**a**) Mean, (**b**) rms, (**c**) maximum misfit, and (**d**) source global conservation for the different pairs of grids for the *harmonic* function for second-order conservative remapping with FRACAREA normalisation. The red circles identify anomalous regriddings for ESMF when the source grid is the icosahedral one (*icos*) detailed in the text.

The only particularity seems to be for ESMF, when the source grid is the icosahedral one (*icos*), which shows a relatively high mean misfit. To better qualify this anomaly, we zoomed in on the 2D representation of the misfit for the *icos-nogt* case, as shown on Figure 15. The misfit shows an alternating positive and negative pattern which causes the relatively high mean misfit for ESMF. Work is underway with ESMF developers to solve this issue.

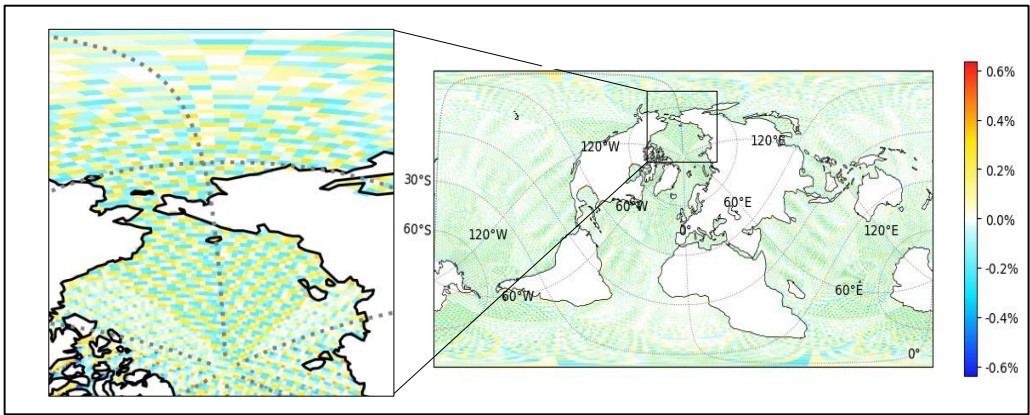

**Figure 15.** Misfit (%) on the target grid *nogt* for the *icos-nogt* second-order conservative remapping with FRACAREA normalisation for ESMF, with a zoom on the left.

Figure 16 shows $L_{min}$ and $L_{max}$ for the second-order conservative remapping with FRACAREA normalisation for the *gulfstream* function, which presents some outstanding values (the other functions do not present such outstanding values). XIOS shows a strong undershoot for *torc-icos*, as shown by $L_{min}$, and a strong overestimate for *bggd-nogt* as shown by $L_{max}$.

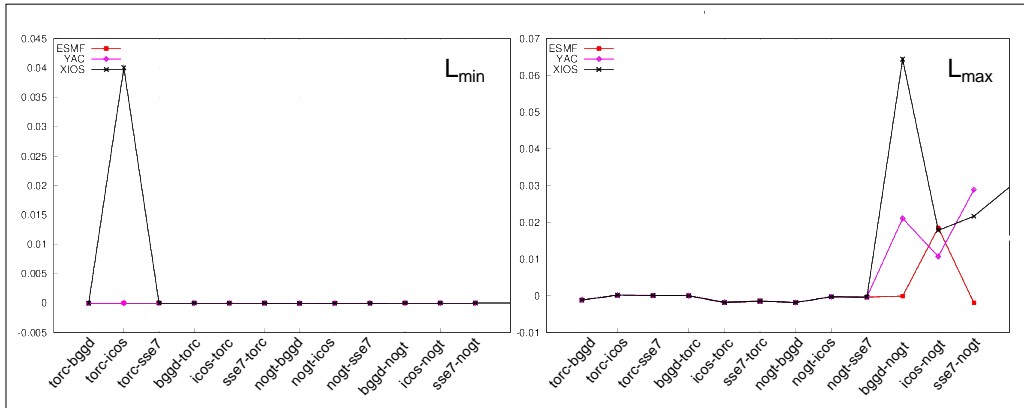

**Figure 16.** $L_{min}$ and $L_{max}$ for the different pairs of grids for the *gulfstream* function for 2nd order conservative remapping with FRACAREA normalisation.

To understand XIOS's undershoot of *torc-icos*, we looked at the 2D misfit in the gulf stream region for XIOS, ESMF and YAC (Figure 17). We observed one clearly outstanding point near the coast for XIOS. ESMF also shows some negative misfit at this point, but it is much smaller than XIOS. YAC does not show any important misfit at this point. This difference between the three regridding libraries has to be investigated in more detail. As they are based on the same algorithm, it must be linked to some implementation differences in the way the libraries calculate the gradients and eventually switch to a 1st order conservative remapping when the gradient cannot be calculated, e.g., near the coast.

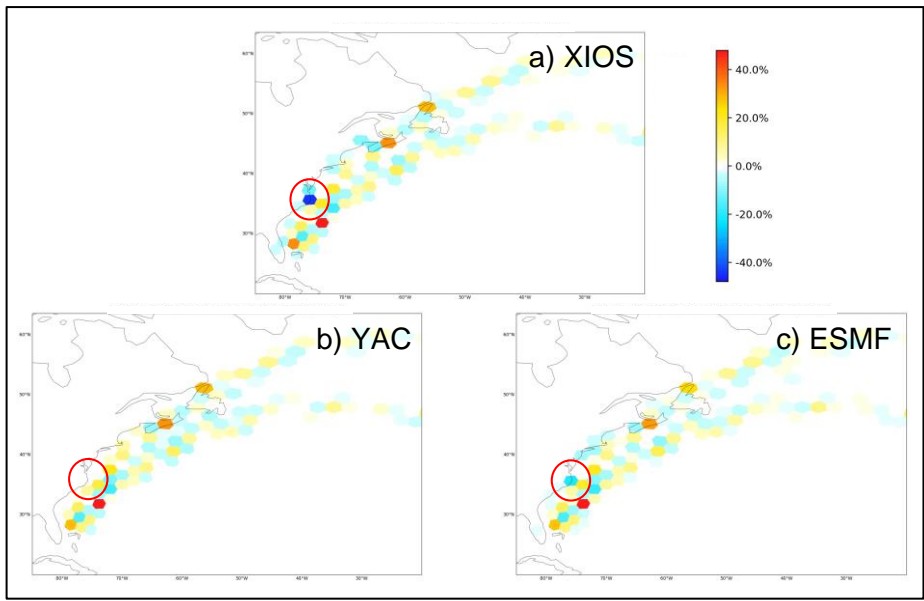

**Figure 17.** Misfit in the gulf stream region for the second-order conservative remapping of the *gulfstream* function for *torc-icos* for XIOS, YAC and ESMF. The red circles identify the grid point near the coast showing an outstanding value for XIOS.

To understand XIOS's overshoot of *bggd-nogt*, we looked at the 2D regridded *gulfstream* function in the gulf stream region for XIOS, ESMF and YAC (not shown). We observed that XIOS shows higher values near the centre of the gulf stream. As for YAC HCSBB (see Figure 8), this behaviour, which explains the overshoot, is most probably linked to some specificities in the algorithm but not to a bug in the implementation.

### 3.7. Comparison of Regridding Algorithms

It is also interesting to compare the results of the different algorithms for each specific library. Figure 18 shows 2D plots of the relative misfit for the remapping of the *vortex* function from the low-resolution icosahedral grid *icos* to the high-resolution icosahedral grid *icoh* with YAC for the (a) first-order conservative remapping and (b) the second-order conservative remapping (both with FRACAREA normalisation). We see the clear benefit of the second order compared to the first order, especially when this remapping involves two grids with very different resolutions. XIOS shows very similar results but not ESMF, probably because of the problem identified above for the second-order conservative remapping for *icos-nogt*, which also exists for *icos-icoh* (alternating positive and negative pattern in the misfit, see Figure 15).

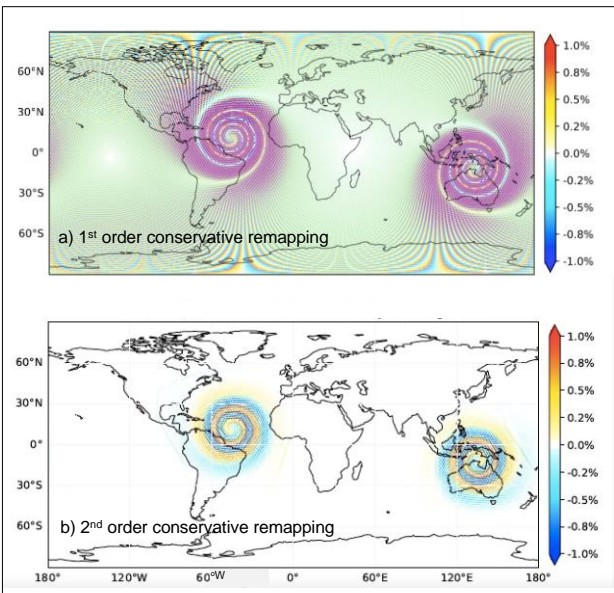

**Figure 18.** Misfit (%) for the (**a**) first-order and (**b**) second-order conservative remapping (both with FRACAREA normalisation) of the *vortex* function from the low-resolution icosahedral grid *icos* to the high-resolution icosahedral grid *icoh* with YAC.

Figure 19 shows the mean misfit and the source global conservation for the different regridding algorithms for ESMF and for YAC. We do not show the equivalent graphs for XIOS as this library supports only conservative regridding, which makes the comparison somewhat limited.

For both ESMF and YAC, Figure 19a,c, respectively, show that the mean misfit for the first-order conservative remapping is always higher than for the second-order remapping. This is expected and fully coherent with the 2D results shown above in Figure 18 for the *icos-icoh* pair of grids.

The comparison of the mean misfit between conservative and non-conservative algorithms does not lead to such clear-cut conclusions. We would expect non-conservative algorithms to show less error at the price of being non-conservative. For ESMF (Figure 19a), we see that this is the case for bilinear and patch when compared with first-order conservative remapping (the green and red curves are under the blue curve for all grid pairs) but their mean misfits are of about the same magnitude as that of the second-order conservative remapping (black curve). For YAC (see Figure 19c), this expectation is basically fulfilled for the HCSBB algorithm (red curve), which shows an error smaller than all of the other algorithms except for the second-order conservative remapping for *torc-icos.* However, the non-conservative first-order regridding (green curve) is the one showing in general the highest error.

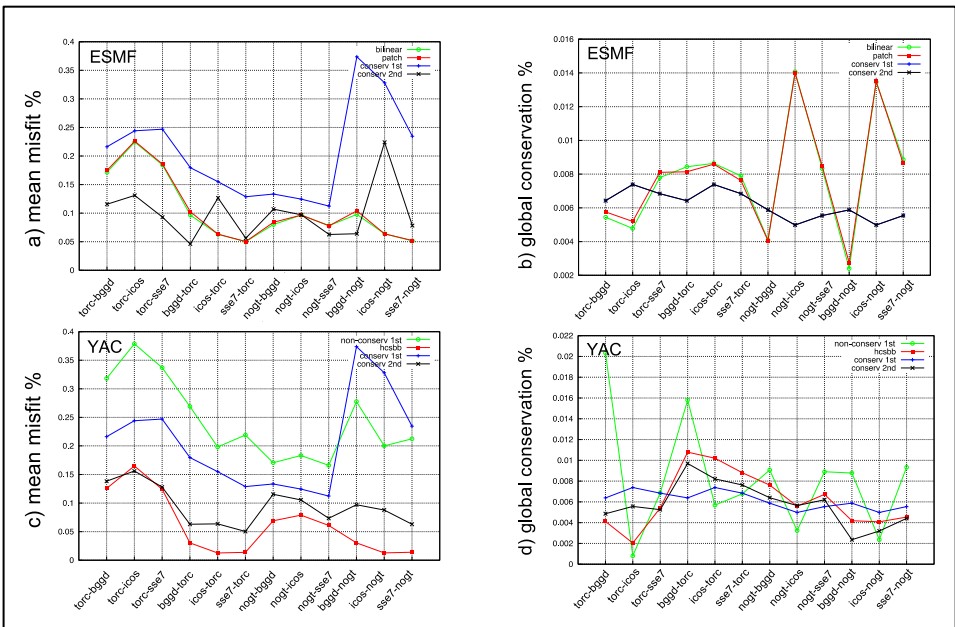

**Figure 19.** Mean misfit (%) for the different regridding algorithms in (**a**) ESMF and (**c**) YAC; source global conservation (%) for the different regridding algorithms in (**b**) ESMF and (**d**) YAC.

Regarding the global conservation (Figure 19b,d), we can observe that the non-conservative remappings (green and red curves) show much more variability with respect to the grid pairs than the first- or second-order remappings. This is reassuring, as it means that the conservative remapping guarantees a certain level of conservation. Still, we observe that for a few pairs of grids, e.g., for *torc-icos* for ESMF, the global conservation is better for non-conservative regriddings than for conservative remappings, which was unexpected a priori.

## 4. Discussion

This paper presents work done to benchmark the quality of four regridding libraries: SCRIP, YAC, ESMF, and XIOS, each evaluating five algorithms (see Section 2.1.3) with four different analytical functions (see Section 2.1.2) for six grids used in real ocean or atmosphere models (see Section 2.1.1).

This benchmark calculates some of the metrics proposed by the CANGA project and we can state that it provides a strong basis to analyse the quality of regridding libraries as it evaluates:

- their sensitivity, as we perform the metric calculation for six pairs of grids in both directions and, in addition, for the *icos-icoh* and *nogt-icoh* pairs for the *vortex* function for second-order conservative FRACAREA remapping;
- their global conservation, as we provide the source and target global conservation metrics.

As we provide and analyse $L_{min}$ and $L_{max}$ metrics, our benchmark also allows a first analysis of the regridding library monotonicity and dissipation (or smoothing). We also started to evaluate the performances of the libraries with a first scalability analysis, not shown here but in [5], that will be completed in the coming months. However, we do not address the regridding library consistency, i.e., the preservation of discretization order and accuracy.

The details of our analysis are the following (note that XIOS offers only first- and second-order conservative remapping):

- ESMF and YAC nearest neighbour regriddings produce almost exactly the same and very reasonable results than the SCRIP (Figure 3). The analytical function defining the field to be regridded has a strong impact on the maximum of the misfit, which is directly linked to the field gradient (Figure 4).
- For first-order non-conservative regridding, YAC, using an inverse distance weighting of the vertex values of the source polygon enclosing the target point, is less accurate

on average than the SCRIP or ESMF bilinear schemes (Figure 5). For second-order non-conservative regridding, the ESMF patch algorithm gives slightly less accurate results than the SCRIP bicubic or the YAC spherical Bernstein–Bézier polynomial algorithms (Figure 6). For first- and second-order non-conservative regridding, all results for ESMF and YAC are reasonable, except for ESMF in the case of the *gulfstream* function for *torc-bggd* and *torc-sse7* grid pairs, which show one anomalous value near the coast (Figure 7). For second-order non-conservative regridding in the case of *icos-torc* and *icos-nogt* for the *gulfstream* function, YAC also shows some higher, but a priori non-anomalous values, in the centre of the gulf stream (Figure 8).

- First-order conservative remapping with DESTAREA normalisation in YAC, ESMF, and XIOS show very similar and good results (Figures 9 and 10), except for ESMF when *nogt* is the source grid if this grid is described with the *SCRIP* (structured) format (Figures 11 and 12). For first-order conservative remapping with FRACAREA normalisation, YAC, ESMF, and XIOS show very similar and good results for all functions (Figure 13); this regridding raises no specific issues for any regridding library.
- YAC, ESMF, and XIOS show approximately the same behaviour with good global conservation for second-order conservative remapping with FRACAREA normalisation, implemented following [21] in the 3 libraries (Figure 14). One issue, however, is in ESMF when the source grid is the icosahedral one, *icos*, which shows a relatively high mean misfit for all functions, with an alternating positive and negative pattern (Figure 15). Another issue is present for XIOS, which shows a strong undershoot for the *gulfstream* function for *torc-icos*, with one clearly outstanding point near the coast.
- The second-order remapping always shows a lower mean misfit than the first-order remapping (Figure 19a,c), and the gain is very evident when going from a low-resolution to a high-resolution grid (Figure 18).
- Unexpectedly, conservative algorithms do not always offer better global conservation than non-conservative ones (Figure 19b,d).

This benchmark leads us to conclude that YAC, ESMF, and XIOS can all three be considered as high-quality regridding libraries, even if some details for few specific cases still need to be fixed. Interactions are currently going on with the library developers to address the very few problems observed.

Benchmarking libraries is always a delicate task as the environment has to be designed to not favour any library a priori. Benchmarking is more than a way to compare libraries and should be taken as a great opportunity for the users to interact with the developers, as we did during the exercise presented here, and for the developers to have their library tested in depth by expert users.

**Author Contributions:** Conceptualization, S.V. and A.P.; data curation, A.P. and G.J.; formal analysis, S.V. and G.J.; funding acquisition, S.V.; investigation, A.P. and G.J.; methodology, S.V. and A.P.; project administration, S.V.; software, A.P. and G.J.; supervision, S.V.; validation, S.V. and G.J.; visualization, S.V., A.P. and G.J.; writing—original draft, S.V. All authors have read and agreed to the published version of the manuscript.

**Funding:** This research was funded by the European Union's Horizon 2020 research and innovation programme in the framework of the IS-ENES3 project under grant agreement No 824084.

**Data Availability Statement:** The OASIS3-MCT sources correspond to the trunk of the OASIS git developer repository dated 5 May 2021. They are available on Zenodo at https://doi.org/10.5281/zenodo.5872502 (accessed on 18 January 2022). The environment used to calculate regridding weights with the SCRIP library in OASIS3-MCT is available in the tar file test_hybrid.tar on Zenodo at https://doi.org/10.5281/zenodo.5342548 (accessed on 31 August 2021). YAC sources correspond to a pre-release state of YAC v2.0.0 that was provided by the developers. All developments used in this version are now included in the official release YAC v2.3.0, available at https://www.dkrz.de/en/services/software-development (accessed on 18 January 2022) and on Zenodo at https://doi.org/10.5281/zenodo.5871066 (accessed on 18 January 2022). The environment to calculate regridding weights with YAC is registered on the CERFACS git developer repository at https://nitrox.cerfacs.fr/globc/OASIS3-MCT/oasis3

-mct_other/tree/master/generate_weights/YAC (accessed on 9 September 2021) and is available on Zenodo at https://doi.org/10.5281/zenodo.5872627 (accessed on 18 January 2022). ESMF sources correspond to the branch ESMF_8_2_0_beta_snapshot_08, which can be obtained with the git command "git clone –branchESMF_8_2_0_beta_snapshot_08–depth 1". They are available on Zenodo at https://doi.org/10.5281/zenodo.5871823 (accessed on 18 January 2022). An environment developed to generate regridding weights with ESMF is available in the tar file generate_weights_ESMF.tar on Zenodo at https://doi.org/10.5281/zenodo.5343048 (accessed on 31 August 2021). XIOS sources correspond to SVN revision 2134 dated 2021-04-29 that can be extracted with the SVN command "svn co -r 2134 http://forge.ipsl.jussieu.fr/ioserver/svn/XIOS/trunk XIOS". They are available on Zenodo at https://doi.org/10.5281/zenodo.5872716 (accessed on 18 January 2022). The environment developed to generate regridding weights with XIOS is available in the tar file generate_weights_XIOS.tar on Zenodo at https://doi.org/10.5281/zenodo.5342491 (accessed on 31 August 2021). The environment used to calculate the benchmark metrics for the four libraries, once the regridding weights were generated for each of them, is available in the tar file compare_interpolation.tar on Zenodo at https://doi.org/10.5281/zenodo.5342778 (accessed on 31 August 2021). The tar file Regridding_Benchmark_metrics.tar, gathering the CSV files containing the benchmark metric values, is available on Zenodo at https://doi.org/10.5281/zenodo.5343166 (accessed on 31 August 2021). The tar file Regridding_Benchmark_metrics_plots.tar, gathering the plots of the regridding benchmark metric, is available on Zenodo at https://doi.org/10.5281/zenodo.5347696 (accessed on 31 August 2021).

**Acknowledgments:** We would like to warmly thank the regridding library developers for their availability and willingness to interact with us during this benchmarking exercise: Moritz Hanke from DKRZ for YAC, Robert Oehmke from NCAR for ESMF and Yann Meurdesoif for CEA/IPSL for XIOS. Interacting with them transformed this benchmarking exercise from a purely analytical work into a community effort, improving regridding libraries used in Earth System Modelling.

**Conflicts of Interest:** The authors declare no conflict of interest. The funders had no role in the design of the study; in the production, analyses, or interpretation of data; in the writing of the manuscript, or in the decision to publish the results.

## Appendix A. Analytical Functions

This appendix contains the definition of the four analytical functions, expressed in Fortran 90, used to define the coupling fields, i.e., *sinusoid* (see Figure A1), *harmonic* (see Figure A2), *vortex* (see Figure A3), *gulfstream* (see Figure A4).

*(A)   sinusoid*

```fortran
!
!! ANALYTICAL SINUSOID
SUBROUTINE function_sinusoid(ni, nj, xcoor, ycoor, fnc_ana)
  !
  IMPLICIT NONE
  !
  INTEGER, PARAMETER :: wp = SELECTED_REAL_KIND(12,307) ! double
  !
  INTEGER, INTENT(IN) :: ni, nj
  REAL(kind=wp), DIMENSION(ni,nj), INTENT(IN)  :: xcoor, ycoor
  REAL(kind=wp), DIMENSION(ni,nj), INTENT(OUT) :: fnc_ana
  !
  REAL (kind=wp), PARAMETER    :: dp_pi=3.14159265359
  REAL (kind=wp), PARAMETER    :: dp_conv = dp_pi/180.
  REAL(kind=wp)  :: dp_length, coef, coefmult
  INTEGER             :: i,j
  !
  DO j=1,nj
    DO i=1,ni
!
      dp_length = 1.2*dp_pi
      coef = 2.
      coefmult = 1.
      fnc_ana(i,j) = coefmult*(coef - COS( dp_pi*(ACOS( COS(xcoor(i,j)*dp_conv)*COS(ycoor(i,j)*dp_conv) )/dp_length)) )
!
    ENDDO
  ENDDO
END SUBROUTINE function_sinusoid
```

**Figure A1.** Fortran 90 code defining the *sinusoid* analytical function.

*(B)* *harmonic*

```fortran
!
!!! HARMONIC FROM TEMPEST-REMAP
SUBROUTINE function_harmonic(ni, nj, xcoor, ycoor, fnc_ana)
 !
 IMPLICIT NONE
 !
 INTEGER, PARAMETER :: wp = SELECTED_REAL_KIND(12,307) ! double
 !
 INTEGER, INTENT(IN) :: ni, nj
 REAL(kind=wp), DIMENSION(ni,nj), INTENT(IN)  :: xcoor, ycoor
 REAL(kind=wp), DIMENSION(ni,nj), INTENT(OUT) :: fnc_ana
 !
 REAL (kind=wp), PARAMETER    :: dp_pi=3.14159265359
 REAL (kind=wp), PARAMETER    :: dp_conv = dp_pi/180.
 !
 INTEGER              :: i,j
 !
 DO j=1,nj
    DO i=1,ni
       fnc_ana(i,j) = 2.0 + SIN( 2.0 * ycoor(i,j)*dp_conv ) ** 16.0  * COS( 16.0 * xcoor(i,j)*dp_conv )
    ENDDO
 ENDDO
END SUBROUTINE function_harmonic
```

**Figure A2.** Fortran 90 code defining the *harmonic* analytical function.

*(C)* *vortex*

```fortran
!
!!! VORTEX FROM TEMPEST-REMAP (as in XIOS)
SUBROUTINE function_vortex(ni, nj, xcoor, ycoor, fnc_ana)
 !
 IMPLICIT NONE
 !
 INTEGER, PARAMETER :: wp = SELECTED_REAL_KIND(12,307) ! double
 !
 INTEGER, INTENT(IN) :: ni, nj
 REAL(kind=wp), DIMENSION(ni,nj), INTENT(IN)  :: xcoor, ycoor
 REAL(kind=wp), DIMENSION(ni,nj), INTENT(OUT) :: fnc_ana
 !
 REAL(kind=wp), PARAMETER :: dp_pi = 3.14159265359
 REAL(kind=wp), PARAMETER :: dLon0 = 5.5
 REAL(kind=wp), PARAMETER :: dLat0 = 0.2
 REAL(kind=wp), PARAMETER :: dR0   = 3.0
 REAL(kind=wp), PARAMETER :: dD    = 5.0
 REAL(kind=wp), PARAMETER :: dT    = 6.0
 REAL(kind=wp) :: dp_length, dp_conv
 !
 REAL(kind=wp) :: dSinC, dCosC, dCosT, dSinT
 REAL(kind=wp) :: dTrm, dX, dY, dZ
 REAL(kind=wp) :: dlon, dlat
 REAL(kind=wp) :: dRho, dVt, dOmega
 !
 INTEGER           :: i,j
 CHARACTER(LEN=7) :: cl_anaftype="vortex"
 !
 dp_conv = dp_pi/180.
 dSinC = SIN( dLat0 )
 dCosC = COS( dLat0 )
 !
 DO j=1,nj
    DO i=1,ni
       ! Find the rotated longitude and latitude of a point on a sphere
       !           with pole at (dLon0, dLat0).
       dCosT = COS( ycoor(i,j)*dp_conv )
       dSinT = SIN( ycoor(i,j)*dp_conv )

       dTrm = dCosT * COS( xcoor(i,j)*dp_conv - dLon0 )
       dX   = dSinC * dTrm - dCosC * dSinT
       dY   = dCosT * SIN( xcoor(i,j)*dp_conv - dLon0 )
       dZ   = dSinC * dSinT + dCosC * dTrm

       dlon = ATAN2( dY, dX )
       IF( dlon < 0.0 ) dlon = dlon + 2.0 * dp_pi
       dlat = ASIN( dZ )

       dRho = dR0 * COS(dlat)
       dVt = 3.0 * SQRT(3.0)/2.0/COSH(dRho)/COSH(dRho)*TANH(dRho)
       IF (dRho == 0.0) THEN
          dOmega = 0.0
       ELSE
          dOmega = dVt / dRho
       END IF

       fnc_ana(i,j) = 2.0 * ( 1.0 + TANH( dRho / dD * SIN( dLon - dOmega * dT ) ) )

    END DO
 END DO
 !
END SUBROUTINE function_vortex
```

**Figure A3.** Fortran 90 code defining the *vortex* analytical function.

*(D) gulfstream*

```fortran
!
!! ANALYTICAL GULF STREAM
SUBROUTINE function_gulfstream(ni, nj, lon, lat, fnc_ana)
!!**********************************************
 !
 INTEGER, PARAMETER :: wp = SELECTED_REAL_KIND(12,307) ! double

 INTEGER, INTENT(IN) :: ni, nj
 REAL (kind=wp), DIMENSION(ni,nj), INTENT(IN)  :: lon, lat
 REAL(kind=wp), DIMENSION(ni,nj), INTENT(OUT) :: fnc_ana

 REAL (kind=wp), PARAMETER :: coef=2., dp_pi=3.14159265359
 REAL (kind=wp) :: dp_length, dp_conv
 INTEGER :: i, j

 ! Analytical Gulf Stream
 REAL (kind=wp) :: gf_coef, gf_ori_lon, gf_ori_lat, &
                 & gf_end_lon, gf_end_lat, gf_dmp_lon, gf_dmp_lat
 REAL (kind=wp) :: gf_per_lon
 REAL (kind=wp) :: dx, dy, dr, dth, dc, dr0, dr1

 ! Parameters for analytical function
 dp_length = 1.2*dp_pi
 dp_conv = dp_pi/180.
 gf_coef = 1.0 ! Coefficient for Gulf Stream term (0.0 = no Gulf Stream)
 gf_ori_lon = -80.0 ! Origin of the Gulf Stream (longitude in deg)
 gf_ori_lat =  25.0 ! Origin of the Gulf Stream (latitude in deg)
 gf_end_lon =  -1.8 ! End point of the Gulf Stream (longitude in deg)
 gf_end_lat =  50.0 ! End point of the Gulf Stream (latitude in deg)
 gf_dmp_lon = -25.5 ! Point of the Gulf Stream decrease (longitude in deg)
 gf_dmp_lat =  55.5 ! Point of the Gulf Stream decrease (latitude in deg)

 dr0 = SQRT(((gf_end_lon - gf_ori_lon)*dp_conv)**2 + &
    & ((gf_end_lat - gf_ori_lat)*dp_conv)**2)
 dr1 = SQRT(((gf_dmp_lon - gf_ori_lon)*dp_conv)**2 + &
    & ((gf_dmp_lat - gf_ori_lat)*dp_conv)**2)

 DO j=1,nj
   DO i=1,ni

     ! Original OASIS fcos analytical test function
     fnc_ana(i,j)=(coef-COS(dp_pi*(ACOS(COS(lat(i,j)*dp_conv)*&
                    & COS(lon(i,j)*dp_conv))/dp_length)))
     gf_per_lon = lon(i,j)
     IF (gf_per_lon > 180.0) gf_per_lon = gf_per_lon - 360.0
     IF (gf_per_lon < -180.0) gf_per_lon = gf_per_lon + 360.0
     dx = (gf_per_lon - gf_ori_lon)*dp_conv
     dy = (lat(i,j) - gf_ori_lat)*dp_conv
     dr = SQRT(dx*dx + dy*dy)
     dth = ATAN2(dy, dx)
     dc = 1.3*gf_coef
     IF (dr > dr0) dc = 0.0
     IF (dr > dr1) dc = dc * COS(dp_pi*0.5*(dr-dr1)/(dr0-dr1))
     fnc_ana(i,j) = fnc_ana(i,j) + &
                    & (MAX(1000.0*SIN(0.4*(0.5*dr+dth) + &
                    &  0.007*COS(50.0*dth) + 0.37*dp_pi),999.0) - 999.0) * dc

   ENDDO
 ENDDO

END SUBROUTINE function_gulfstream
```

**Figure A4.** Fortran 90 code defining the *gulfstream* analytical function.

## Appendix B. List of CSV Files Containing Metrics Values

The files included in the tar file Regridding_Benchmark_metrics.tar available on Zenodo at https://doi.org/10.5281/zenodo.5343166 (accessed on 31 August 2021) are listed here below in Table A1. This tar file contains the regridding benchmark metrics calculated for all pairs of grids (see Section 2.1.1) and all functions (see Section 2.1.2) for all regridding libraries SCRIP (+SCRIP-L, i.e., with Lambert projection for conservative regridding), YAC, ESMF, and XIOS, and for all algorithms (see Section 2.1.3), except in cases where the regridding library does not support the algorithm (e.g., nearest neighbour for XIOS).

The name of the file is given as R_A_f.csv, where R is the regridding library, A is the algorithm, and f is the function (here *classic* is equivalent to *sinusoid*). The algorithm A can be:

- "DISTWGT_1" for nearest neighbour
- "BILINEAR" for first-order non conservative
- "BICUBIC" for second-order non-conservative
- "CONSERV" for first-order conservative
- "CONS2ND" for second-order conservative

For conservative remapping, A also contains the normalisation option "FRACAREA" or "DESTAREA".

For XIOS, there are therefore no files for nearest neighbour, first- and second-order non-conservative algorithms as they are not supported in XIOS. For the first- and second-order conservative remapping for SCRIP, there are two files: one with (SCRIP-L) and one without (SCRIP) the Lambert azimuthal projection. For ESMF, all results are provided for the version tagged ESMF_8_2_0_beta_snapshot_08 (ESMF-820bs08). For ESMF, for first- and second-order conservative algorithms, the *nogt* grid was described as *unstructured*, as it correctly supports the north fold of the NEMO grid (see Section 3.4), in that case, R = "ESMF-820bs08-U". For first-order conservative with DESTAREA normalisation, results are also provided that describe the *nogt* grid as with the *SCRIP* format for comparison, in that case, R = "ESMF-820bs08".

**Table A1.** Regridding benchmark files included in the tar file Regridding_Benchmark_metrics.tar available on Zenodo at https://doi.org/10.5281/zenodo.5343166 (accessed on 31 August 2021).

| | |
|---|---|
| ESMF-820bs08-U_CONS2ND_FRACAREA_classic.csv | SCRIP_CONS2ND_FRACAREA_harmonic.csv |
| ESMF-820bs08-U_CONS2ND_FRACAREA_gulfstream.csv | SCRIP_CONS2ND_FRACAREA_vortex.csv |
| ESMF-820bs08-U_CONS2ND_FRACAREA_harmonic.csv | SCRIP_CONSERV_DESTAREA_classic.csv |
| ESMF-820bs08-U_CONS2ND_FRACAREA_vortex.csv | SCRIP_CONSERV_DESTAREA_gulfstream.csv |
| ESMF-820bs08-U_CONSERV_DESTAREA_classic.csv | SCRIP_CONSERV_DESTAREA_harmonic.csv |
| ESMF-820bs08-U_CONSERV_DESTAREA_gulfstream.csv | SCRIP_CONSERV_DESTAREA_vortex.csv |
| ESMF-820bs08-U_CONSERV_DESTAREA_harmonic.csv | SCRIP_CONSERV_FRACAREA_classic.csv |
| ESMF-820bs08-U_CONSERV_DESTAREA_vortex.csv | SCRIP_CONSERV_FRACAREA_gulfstream.csv |
| ESMF-820bs08-U_CONSERV_FRACAREA_classic.csv | SCRIP_CONSERV_FRACAREA_harmonic.csv |
| ESMF-820bs08-U_CONSERV_FRACAREA_gulfstream.csv | SCRIP_CONSERV_FRACAREA_vortex.csv |
| ESMF-820bs08-U_CONSERV_FRACAREA_harmonic.csv | SCRIP_DISTWGT_1_classic.csv |
| ESMF-820bs08-U_CONSERV_FRACAREA_vortex.csv | SCRIP_DISTWGT_1_gulfstream.csv |
| ESMF-820bs08_BICUBIC_classic.csv | SCRIP_DISTWGT_1_harmonic.csv |
| ESMF-820bs08_BICUBIC_gulfstream.csv | SCRIP_DISTWGT_1_vortex.csv |
| ESMF-820bs08_BICUBIC_harmonic.csv | XIOS_CONS2ND_FRACAREA_classic.csv |
| ESMF-820bs08_BICUBIC_vortex.csv | XIOS_CONS2ND_FRACAREA_gulfstream.csv |
| ESMF-820bs08_BILINEAR_classic.csv | XIOS_CONS2ND_FRACAREA_harmonic.csv |
| ESMF-820bs08_BILINEAR_gulfstream.csv | XIOS_CONS2ND_FRACAREA_vortex.csv |
| ESMF-820bs08_BILINEAR_harmonic.csv | XIOS_CONSERV_DESTAREA_classic.csv |
| ESMF-820bs08_BILINEAR_vortex.csv | XIOS_CONSERV_DESTAREA_gulfstream.csv |
| ESMF-820bs08_CONSERV_DESTAREA_classic.csv | XIOS_CONSERV_DESTAREA_harmonic.csv |
| ESMF-820bs08_CONSERV_DESTAREA_gulfstream.csv | XIOS_CONSERV_DESTAREA_vortex.csv |
| ESMF-820bs08_CONSERV_DESTAREA_harmonic.csv | XIOS_CONSERV_FRACAREA_classic.csv |
| ESMF-820bs08_CONSERV_DESTAREA_vortex.csv | XIOS_CONSERV_FRACAREA_gulfstream.csv |
| ESMF-820bs08_DISTWGT_1_classic.csv | XIOS_CONSERV_FRACAREA_harmonic.csv |
| ESMF-820bs08_DISTWGT_1_gulfstream.csv | XIOS_CONSERV_FRACAREA_vortex.csv |
| ESMF-820bs08_DISTWGT_1_harmonic.csv | YAC_BICUBIC_classic.csv |
| ESMF-820bs08_DISTWGT_1_vortex.csv | YAC_BICUBIC_gulfstream.csv |
| SCRIP-L_CONS2ND_FRACAREA_classic.csv | YAC_BICUBIC_harmonic.csv |
| SCRIP-L_CONS2ND_FRACAREA_gulfstream.csv | YAC_BICUBIC_vortex.csv |
| SCRIP-L_CONS2ND_FRACAREA_harmonic.csv | YAC_BILINEAR_classic.csv |
| SCRIP-L_CONS2ND_FRACAREA_vortex.csv | YAC_BILINEAR_gulfstream.csv |
| SCRIP-L_CONSERV_DESTAREA_classic.csv | YAC_BILINEAR_harmonic.csv |
| SCRIP-L_CONSERV_DESTAREA_gulfstream.csv | YAC_BILINEAR_vortex.csv |
| SCRIP-L_CONSERV_DESTAREA_harmonic.csv | YAC_CONS2ND_FRACAREA_classic.csv |

**Table A1.** *Cont.*

| | |
|---|---|
| SCRIP-L_CONSERV_DESTAREA_vortex.csv | YAC_CONS2ND_FRACAREA_gulfstream.csv |
| SCRIP-L_CONSERV_FRACAREA_classic.csv | YAC_CONS2ND_FRACAREA_harmonic.csv |
| SCRIP-L_CONSERV_FRACAREA_gulfstream.csv | YAC_CONS2ND_FRACAREA_vortex.csv |
| SCRIP-L_CONSERV_FRACAREA_harmonic.csv | YAC_CONSERV_DESTAREA_classic.csv |
| SCRIP-L_CONSERV_FRACAREA_vortex.csv | YAC_CONSERV_DESTAREA_gulfstream.csv |
| SCRIP_BICUBIC_classic.csv | YAC_CONSERV_DESTAREA_harmonic.csv |
| SCRIP_BICUBIC_gulfstream.csv | YAC_CONSERV_DESTAREA_vortex.csv |
| SCRIP_BICUBIC_harmonic.csv | YAC_CONSERV_FRACAREA_classic.csv |
| SCRIP_BICUBIC_vortex.csv | YAC_CONSERV_FRACAREA_gulfstream.csv |
| SCRIP_BILINEAR_classic.csv | YAC_CONSERV_FRACAREA_harmonic.csv |
| SCRIP_BILINEAR_gulfstream.csv | YAC_CONSERV_FRACAREA_vortex.csv |
| SCRIP_BILINEAR_harmonic.csv | YAC_DISTWGT_1_classic.csv |
| SCRIP_BILINEAR_vortex.csv | YAC_DISTWGT_1_gulfstream.csv |
| SCRIP_CONS2ND_FRACAREA_classic.csv | YAC_DISTWGT_1_harmonic.csv |
| SCRIP_CONS2ND_FRACAREA_gulfstream.csv | YAC_DISTWGT_1_vortex.csv |

## Appendix C. List of Metric Plots

The plots included in the tar file Regridding_Benchmark_metrics_plots.tar available on Zenodo at https://doi.org/10.5281/zenodo.5347696 (accessed on 31 August 2021) are listed here below in Table A2.

This tar file contains the regridding benchmark metric plots calculated for all pairs of grids (see Section 2.1.1), for all functions (see Section 2.1.2), for all regridding libraries (SCRIP, SCRIP-L, i.e., with Lambert projection for conservative regridding, YAC, ESMF, and XIOS) and for all algorithms (see Section 2.1.3), except when the regridding library does not support the algorithm (e.g., bilinear for XIOS).

There is one plot for each algorithm, for each function, for all metrics, for all pairs of grids, and for all regridding libraries. The name of the file is given as plot_remap_metrics_A_f.pdf, where A is the algorithm (see Appendix B) and f is the function (here *classic* is equivalent to *sinusoid*).

For XIOS, there is no plot for nearest neighbour, first- and second-order non-conservative algorithms as they are not supported by XIOS.

For ESMF, all plots are provided for the version tagged ESMF_8_2_0_beta_snapshot_08 with *nogt* described with the *unstructured* format, as it correctly supports the north fold of the NEMO grid (see Section 3.4).

**Table A2.** Regridding benchmark plots included in the tar file Regridding_Benchmark_metrics_plots.tar available on Zenodo at https://doi.org/10.5281/zenodo.5347696 (accessed on 31 August 2021).

| |
|---|
| plot_remap_metrics_BICUBIC_classic.pdf |
| plot_remap_metrics_BICUBIC_gulfstream.pdf |
| plot_remap_metrics_BICUBIC_harmonic.pdf |
| plot_remap_metrics_BICUBIC_vortex.pdf |
| plot_remap_metrics_BILINEAR_classic.pdf |
| plot_remap_metrics_BILINEAR_gulfstream.pdf |
| plot_remap_metrics_BILINEAR_harmonic.pdf |
| plot_remap_metrics_BILINEAR_vortex.pdf |
| plot_remap_metrics_CONS2ND_FRACAREA_classic.pdf |
| plot_remap_metrics_CONS2ND_FRACAREA_gulfstream.pdf |
| plot_remap_metrics_CONS2ND_FRACAREA_harmonic.pdf |
| plot_remap_metrics_CONS2ND_FRACAREA_vortex.pdf |
| plot_remap_metrics_CONSERV_DESTAREA_classic.pdf |
| plot_remap_metrics_CONSERV_DESTAREA_gulfstream.pdf |
| plot_remap_metrics_CONSERV_DESTAREA_harmonic.pdf |
| plot_remap_metrics_CONSERV_DESTAREA_vortex.pdf |
| plot_remap_metrics_CONSERV_FRACAREA_classic.pdf |
| plot_remap_metrics_CONSERV_FRACAREA_classic_sansconservationMTR.pdf |

**Table A2.** *Cont.*

| |
| --- |
| plot_remap_metrics_CONSERV_FRACAREA_gulfstream.pdf |
| plot_remap_metrics_CONSERV_FRACAREA_harmonic.pdf |
| plot_remap_metrics_CONSERV_FRACAREA_vortex.pdf |
| plot_remap_metrics_DISTWGT_1_classic.pdf |
| plot_remap_metrics_DISTWGT_1_gulfstream.pdf |
| plot_remap_metrics_DISTWGT_1_harmonic.pdf |
| plot_remap_metrics_DISTWGT_1_vortex.pdf |

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
