# Peer review of "Benchmarking Regridding Libraries Used in Earth System Modelling"

_mca, doi:10.3390/mca27020031_

Round 1

Reviewer 1 Report

Please find my comments and suggestions for the authors in the attached PDF.

Reviewer 2 Report

This is a nice manuscript presenting the work done to benchmark the quality of four regridding libraries commonly used in ESMs. This work is very useful to ESM’s developers. Overall, it is clearly written and well organized. The manuscript gives a comprehensive and useful introduction to the four regridding libraries. This manuscript should be acceptable for publication in MCA after minor changes mentioned below.

General comments

  1. In section 2.2, the compiler version used to calculate the regridding weights in different libraries should be described. Could you give an example to show the differences of regridding weights generated in different platforms?

  1. When running coupled model, some coupling fields, such as sea surface temperature, with missing value over land need to regrid from ocean model grids to atmosphere model grids. In this case, the anomalous values may be occurred near the coast after regridding. Could you address this case in your paper?

  1. What do you think about the vertical interpolation? Do you plan to benchmark the vertical regridding libraries in the future?

Specific comments.

  1. Line 13: I think “ECMWF” should be “ESMF”.

  1. Line 126: What’s “CANGA” stands for? The full name should be given.

  1. I recommend you give the schematic figures of different grids mentioned in section 2.1.1.

  1. Figure 16: The location of the red circles in three panels are not the same.
